# Effects of Ibuprofen Intake in Muscle Damage, Body Temperature and Muscle Power in Paralympic Powerlifting Athletes

**DOI:** 10.3390/ijerph17145157

**Published:** 2020-07-17

**Authors:** Guacira S. Fraga, Felipe J. Aidar, Dihogo G. Matos, Anderson C. Marçal, Jymmys L. Santos, Raphael F. Souza, André L. Carneiro, Alan B. Vasconcelos, Marzo E. Da Silva-Grigoletto, Roland van den Tillaar, Breno T. Cabral, Victor M. Reis

**Affiliations:** 1Department of Physical Education, Federal University of Sergipe (UFS), São Cristovão, Sergipe 49100-000, Brazil; guacirafraga@yahoo.com.br (G.S.F.); fjaidar@gmail.com (F.J.A.); acmarcal@yahoo.com.br (A.C.M.); desouza@ufs.br (R.F.S.); abs.vasconcelos@gmail.com (A.B.V.); dasilvame@gmail.com (M.E.D.S.-G.); 2Group of Studies and Research of Performance, Sport, Health and Paralympic Sports (GEPEPS), Federal University of Sergipe (UFS), São Cristovão, Sergipe 49100-000, Brazil; dihogogmc@gmail.com; 3Northeast Network in Biotechnology (RENORBIO), Federal University of Sergipe, Sergipe 49100-000, Brazil; jymmys.lopes@gmail.com; 4Department of Physical Education, State University of Montes Claros, Minas Gerais 30000-000, Brazil; algcarneiro@gmail.com; 5Department of Sport Sciences and Physical Education, Nord University, 7600 Levanger, Norway; roland.v.tillaar@nord.no; 6Department of Physical Education, Federal University of Rio Grande do Norte, Natal 59072970, Brazil; brenotcabral@gmail.com; 7Research Center in Sports Sciences, Health Sciences and Human Development (CIDESD), University of Trás-os-Montes and Alto Douro, 5000-412 Vila Real, Portugal

**Keywords:** paralympic powerlifting, ibuprofen, muscle function, muscle damage

## Abstract

The aim of this study is to evaluate the effect of ingesting ibuprofen on post-workout recovery of muscle damage, body temperature and muscle power indicators in Paralympic powerlifting athletes. The study was carried out with eight Paralympic powerlifting athletes (aged 27.0 ± 5.3 years and 79.9 ± 25.5 kg of body mass) competing at the national level, with a minimum training experience of 12 months, who all submitted to two experimental conditions: Ibuprofen (2 × 00 mg) and control. The maximal isometric force of the upper limbs and rate of force development, thermography, and serum biochemical analyzes of creatine kinase, lactate dehydrogenase, aspartate aminotransferase and alanine aminotransferase were measured before, after, 24 h after and 48 h after the intervention. Maximal isometric force only decreased in the placebo condition, which increased back to baseline levels, while no substantial decline in baseline force was seen in the ibuprofen condition, although no effect for exercise condition was detected. After the exercise, the rate of force development decreased significantly for both conditions and did not exceed baseline levels again after 48 h. Muscle temperature decreased significantly at 48-h post-exercise in the placebo condition, when compared with the previous day of measurement; and deltoid muscle temperature at 48-h post-exercise was higher with the ibuprofen condition. Although the results indicate some positive effects of ibuprofen use, they do not enable a clear statement regarding its positive effects on muscle function and muscle damage. Ibuprofen seems to have caused a delay in the anti-inflammatory response following exercise.

## 1. Introduction

Training recovery is important, as it ensures that training and success can continue at high intensities and longer durations to further stimulate the body and trigger adaptations [1]. Athletes practice several days a week in elite sports, and several times a day, and if the recovery is not complete, the workload of the training must be decreased [1,2].

The appearance of inflammation, soreness, and the decrease in athletes´ performance are associated with exercise-induced muscle damage. It is due to biochemical stress (oxidative stress) and mechanical stress (contractions) that disrupts the muscle cell membrane and damage the muscle fibers’ Z bands [3]. This exercise-induced muscle damage has long been quantified using skeletal muscle protein and enzyme serum levels (creatine kinase (CK), troponin I, and myoglobin) as markers [3]. To speed up recovery processes and prevent muscle soreness, athletes have used various techniques, such as cryotherapy and non-steroidal anti-inflammatory drugs (NSAIDs) (i.e., ibuprofen), both during competition and training [4,5]. Some of these techniques have been used based on scientifically hypothesized mechanisms, but the results have been inconsistent. Some have reported that ibuprofen had no major effects on inflammatory response following resistance exercise [6,7], whereas others [8] supported its benefits in lowering CK levels and reporting lower muscle soreness after eccentric leg curl exercises, though without benefits in restoring muscle function. A recent metanalysis supports NSAID use as a means to lower strength loss, soreness, and blood creatine kinase level after an acute muscle injury [9] and ibuprofen has been shown to mitigate fatigue in competitive male runners [10].

Although NSAIDs are often used to alleviate exercise-induced muscle soreness and speed up the recovery process after exercise, there is no support for such an effect on Paralympic powerlifting athletes to date. A potential benefit could come from ibuprofen use during very intense training short-periods (i.e., a week), where athletes seek to accumulate an unusual amount of work. Therefore, this study aims to analyze the effect of ibuprofen on post-workout recovery in Paralympic powerlifting athletes. Specifically, this study investigates the potential effect of ibuprofen ingestion to reduce muscle damage, body temperature, and preserve muscle power. It is hypothesized that the ibuprofen would enhance recovery from powerlifting, as well as reduce muscle damage, body temperature, and preserve muscle power.

## 2. Materials and Methods

### 2.1. Participants

The present study consisted of a randomized cross-over clinical trial. Eight Paralympic powerlifting athletes volunteered for the study (age = 27.0 ± 5.3 years, body mass = 79.9 ± 25.3 kg, training experience = 3.5 ± 0.2 years, 1RM bench press = 133.1 ± 31.4 kg, 1RM/body mass = 1.7 ± 0.3). The athletes fulfilled all prerequisites of the Brazilian Paralympic Committee [11], and were classified as national elite athletes. Two athletes had a spinal cord injury below the eighth thoracic vertebra, one had sequelae due to poliomyelitis, one had cerebral palsy, two had a malformation of the lower limbs, and two were amputated. Athletes were not included in the study if they (1) reported the consumption of banned substances, (2) had been previously diagnosed with cardiac or metabolic disease, or (3) were involved in any process to induce rapid weight loss at the time of recruitment. The athletes participated voluntarily and signed an informed consent form in accordance with Resolution 466/2012 of the National Commission for Research Ethics (CONEP) of the National Health Council, and the ethical principles of the latest version of the Declaration of Helsinki 2013 (and the World Medical Association). This study was approved by the Research Ethics Committee of the Federal University of Sergipe, CAAE: 79909917.0.0000.55.46.

### 2.2. Experimental Design

All athletes underwent a 1 repetition maximal (1RM) bench press test twice, with a 48 h rest between tests. After a standardized warm-up, each subject started the attempts with a weight that he believed could be lifted only once using maximum effort. Increases in weight were added until the maximum load that could be lifted once was reached. If the athlete failed to perform a single repetition, 2.5% of the load used in the test were subtracted [12]. The subjects rested for 3–5 min between attempts. The largest record between the two sessions was taken as the individual´s 1RM. Coefficient of variation between the two measures was ICC > 93%.

On the subsequent two weeks, participants underwent a training session with bench press exercises with one week in between. All participants randomly trained using the two different recovery methods: ingesting placebo (PLA) or ibuprofen (IBU). All assessments were carried out 30 min before the training started, immediately at the end, 24 h and 48 h after the training (Table 1). Assessments included: (i) Measurement of muscle function; (ii) thermography; and (iii) blood collections.

The intervention protocol consisted of warm-up for upper limbs, using three exercises (abduction of the shoulders with dumbbells, elbow extension in the pulley and rotation of the shoulders with dumbbells) with three sets of 10 to 20 repetitions [13]. Soon after, a specific warm-up was performed on the bench press with a 30% load of 1RM, 10 slow repetitions (3:1 s, eccentric: concentric) and 10 fast repetitions (1:1 s, eccentric: concentric). This was followed with five sets of bench press of five maximum repetitions (5 sets—85 at 90% RM), using a fixed load. The complete session lasted for 1 h 30 min. During the test, athletes received verbal encouragement to achieve maximum performance [13]. To perform the bench press, an official straight bench (Eleiko, Chicago, IL, USA), approved by the International Paralympic Committee [11] was used.

Ingestion of IBU (ibuprofen) or PLA (placebo) occurred 15 min before and 5 h post-training, according to De Souza et al. [10]. Participants received two capsules of IBU (each capsule containing 400 mg) and were instructed to ingest one capsule before training and one capsule post-training. In the control condition, two flour capsules were delivered. Both IBU and PLA were packaged in identical capsules. The experiment was double-blind, and the order of distribution of the capsules was determined at random.

### 2.3. Measurements

The evaluations of maximal isometric force and rate of force development (RFD) were determined by a Muscle Lab load cell (Model PFMA 3010 and Muscle Lab System; Ergotest, Langesund, Norway), attached to the bench by Spider HMS carabiners (Simond, Chamonix, France) and a steel chain with breaking loads of 21 kg and 2300 kg, respectively. The perpendicular distance between the load cell and the center of the joint was determined to calculate the joint torque [14]. Each subject performed three repetitions of 5 s of maximal effort with 10min of rest between repetitions to measure maximal isometric force [15]. A steel chain was used to fix the load cell to the bench. The perpendicular distance between the load cell and the center of the joint was adjusted so that there was approximately 90° of elbow angle, and 15 cm from the bar to the external bone. The RFD was determined using the force to time ratio until reaching the maximal force (RFD = ΔStrength/ΔTime) [14,16]. This distance was determined by measuring the range of motion with a goniometer (FL6010, Sanny, São Paulo, Brazil).

Image captures were performed by an infrared camera model FLIR T640sc (Flir, Stockholm, Sweden) measuring range −40 °C to 2000 °C, accuracy 2%, sensitivity < 0.035, infrared spectral band of 7.5–14 μm, refresh rate of 30 Hz, resolution of 640 × 480 pixels. The software used for thermal image analysis was FLIR TOOLS (Flir, Stockholm, Sweden). The procedures were performed according to the recommendations of the European Association of Thermology [17]. The temperature and relative humidity were controlled between 21–22 °C and 42–50% monitored by a Hybrid Thermo-Hygrometer Hikari HTH-240 (Hikari, Shenzhen, China). The following body regions were photographed: chest and arm, aiming to verify the activities of the Pectoralis Major Sternal portion and the Deltoid Anterior portion.

Serum concentrations of blood variables of CK (creatine kinase), LDH (lactate dehydrogenase), AST (aspartate aminotransferase) and ALT (alanine aminotransferase) were determined using commercial kits (LabTest^®^, Minas Gerais, Brazil), analyzed in a spectrophotometer (Model Bioespectro SP-22 UV/Visible, Minas Gerais, Brazil) under the wavelength of 340 nm.

A digital scale (Toledo, São Paulo, Brazil), with a capacity of 0 to 150 kg and an accuracy of 0.05 kg, was used to determine body mass measurements. The height of the athletes was not evaluated due to the existence of different physical disabilities.

### 2.4. Statistical Analysis

Data are presented as means and standard deviations. Data are checked for normal distribution using the Shapiro Wilk test. To assess the effect of ibuprofen upon the different variables a 2 (condition: placebo, ibuprofen) × 4 (test time: before, after 24 h and 48 h) analysis of variance with repeated measures was performed. If significant effects were found for testing time, a Holm-Bonferroni’s Post Hoc test was conducted. If *p*-values for sphericity (Mauchly’s test) assumptions were violated, corrections with Greenhouse-Geisser were considered. The level of significance was set at *p* < 0.05. The analysis was performed with SPSS Statistics for Windows, version 25.0 (IBM Corp., Armonk, NY, USA). Effect size was evaluated with partial eta squared (η^2^) where 0.01 < η^2^ < 0.06 constituted a small effect, 0.06 < η^2^ < 0.14 a medium effect, and η^2^ > 0.14 a large effect [18].

## 3. Results 

A significant effect of testing time was found for all variables (F ≥ 4.6; *p* ≤ 0.012; η² ≥ 0.40) except triceps temperature and lactate dehydrogenase concentration (F ≤ 1.5; *p* ≥ 0.24; η² ≤ 0.18). Only a significant effect of ibuprofen was found for creatine kinase (F = 11.2; *p* = 0.012; η² = 0.6). However, also a significant test time × condition interaction was found for deltoid temperature (F = 10.3; *p* < 0.001; η² = 0.59) and triceps temperature (F = 3.3; *p* = 0.039; η² = 0.32). Post hoc comparison revealed that maximal isometric force significantly decreased over time in the placebo condition, with subsequent return to baseline levels. In the ibuprofen condition, no significant decrease from baseline was found in force, and there was a subsequent significant increase between 24 and 48 h. The rate of force development decreased significantly after the exercise for both conditions and did not reach baseline levels again after 48 h (Figure 1).

Creatine kinase concentration levels increased significantly at 24 h after the exercise after which they decreased again. The concentration levels increased significantly 24 h post-exercise compared with baseline after placebo ingestion. Aspartate aminotransferase concentration levels increased immediately after the work out with the placebo protocol, while in the ibuprofen protocol it increased after 24 h. Alanine aminotransferase concentration levels increased directly after both ingestion protocols. However, after 48 h a significantly lower concentration level after the ibuprofen ingestion protocol was found compared with the placebo protocol (Figure 2).

Muscle temperature decreased significantly from baseline to 24 h after the placebo intervention for the deltoid muscle and from 24 to 48 h for the deltoid and triceps muscles. The temperature rose significantly at 48-h post-exercise in the ibuprofen condition, when compared with the previous day of measurement; and deltoid muscle temperature at 48-h post-exercise was higher with the ibuprofen condition compared with placebo. Significant differences in triceps muscles at 24 h and deltoid muscle, 48 h between the two conditions were observed, with higher mean values in the ibuprofen condition. (Figure 3).

## 4. Discussion

The present study aimed to analyze the effect of ibuprofen on post-workout recovery in Paralympic powerlifting athletes, thereby checking the potential effect of ibuprofen ingestion to reduce muscle damage, body temperature and preserve muscle power.

Maximal isometric force decreased in the placebo condition, while in the ibuprofen group, it was found a significant increase between 24 h and 48 h. These results match those in the study by Souza et al. [10], where ibuprofen has been shown to mitigate fatigue in competitive male runners, as given by a better performance in the squat jump after the race, when compared with the control group. Contrarily, Correa et al. [19] did not find significant differences in muscle function of individuals who made prophylactic use of ibuprofen (1.2 g) before resistance exercise (bench press and squat, 65% of 1RM), as given by the number of repetitions performed during the training session itself. However, no clear effect of the ibuprofen condition was detected herein.

The assessment of RFD has been used for strength diagnosis, to monitor the effects of training interventions in both healthy populations and patients, discriminate high-level athletes from those of lower levels, evaluate the impairment in mechanical muscle function after acute bouts of eccentric muscle actions and estimate the degree of fatigue and recovery after acute exhausting exercise [20,21]. The rate of force development (RFD) is obtained through the ratio between the variation of force and variation of time, with the maximum values being reached in a period of time between 100 and 300 ms [20,21]. Therefore, RFD has been considered an important criterion to measure neuromuscular performance in athletes who participate in modalities that involve explosive muscle contractions [22]. In the present study, RFD was reduced after exercise in both groups, and it did not return to baseline levels for 48-h. Nevertheless, the evaluation of RFD in human skeletal muscle is a complex task as influenced by numerous distinct methodological factors, including mode of contraction, type of instruction, the method used to quantify RFD, devices used for force/torque recording, and ambient temperature [23].

Among the various muscle damage blood markers that were assessed, CK was higher in the placebo condition when compared with the ibuprofen group at 24 h post-exercise. Post-exercise Alanine aminotransferase increased similarly in both groups, but it was lower in the ibuprofen group at 48 h post-exercise. Souza et al. [10] also investigated these blood markers in runners who were supplemented with 800 mg of ibuprofen before a competitive race. They observed CK, LDH, and AST increased values post-exercise, despite the supplementation. It is not possible to confront these with the results herein, as we have now measured the biomarkers for 48 h post-exercise. Indeed, we found that after 48h, the body begins to balance the markers of muscle damage, irrespective of the ibuprofen use. This is in agreement with previous observations in ultramarathon runners, where markers of muscle/liver injury suffer alterations during exercise, but they are generally transient and reversible [24]. The literature yields contradictory results, though a recent metanalysis showed positive effects of NSAIDs after muscle injury [9]. Whether these effects are also true in non-injury conditions, as those in the present study, remains unclear. The fact that CK was lower in the ibuprofen condition compared with the placebo condition at baseline and that no interaction condition x time was detected, does not enable us to conclude on the effects of the ibuprofen use.

The mechanism of action by which non-steroidal anti-inflammatory drugs (i.e., ibuprofen) relieve pain and is possibly found by inhibiting the synthesis of prostaglandins, endogenous intermediary substances in the inflammatory action, by inactivating two isoenzymes, the constitutive cyclooxygenase (COX-1) and inducible cyclooxygenase (COX-2) [25]. Moreover, prostaglandins are one of the mechanisms that may explain an elevated body temperature. The increase in body temperature, due to a response to physical stress, implies many metabolic changes [26]. In the present study, we also included infrared thermography to assess the possible influence of ibuprofen in post-exercise upper-body temperature. In fact, a higher temperature is expected to be found after intensive anaerobic training when compared with low intensity aerobic training [27].

The post-exercise temperature in the upper-body was not clearly affected by the ibuprofen condition. Though, a lower triceps temperature with ibuprofen was found at 24 h. However, at the same time of measurement (24 h) and also at 48 h post-exercise, the deltoid temperature was higher in the ibuprofen condition. Temperature increases at 24-h and 48-h post-exercise may reflect, at least in the ibuprofen condition, a delay in the anti-inflammatory response to resistance exercise. We have not extended the measurements after 48 h post-exercise, though the process of muscle recovery tends to increase the body temperature at 48, 72, and even 96 h post-exercise, especially after high-intensity exercise [28]. It is possible that muscle baseline temperature may have been influenced by non-controlled factors, such as food intake. In addition, the baseline upper-body temperature measurement in individuals that use wheelchairs is difficult. Indeed, as patients use their wheelchair for every small locomotion that is warranted during the experiment, a true resting baseline may be hard to attain.

Despite some limitations of the present study (sample size, short-period of the experiment and pre-training nutritional intake), the fact that it was performed in a very special population may serve as a basis for future studies on powerlifters. We suggest the inclusion of extra measurements across a prolonged period of time.

## 5. Conclusions

The intake of non-steroidal anti-inflammatory drugs (NSAIDs), such as ibuprofen, has been used by athletes to relieve the symptoms of muscle injury and pain. Our results show that although there are some positive effects of ibuprofen use, there is no clear indication that ibuprofen use has a positive effect on muscle function and muscle damage. Furthermore, the effect of ibuprofen to prevent the normal decrease in muscle temperature during post-exercise recovery may potentially be indicative of a delay in the anti-inflammatory response.

## Figures and Tables

**Figure 1 ijerph-17-05157-f001:**
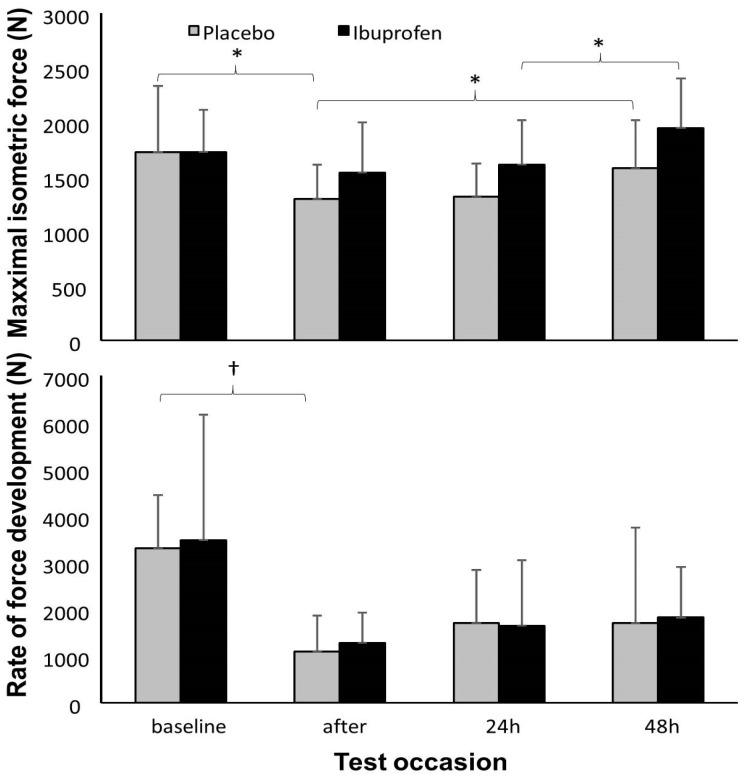
Mean (±SD) maximal isometric force and rate of force development before, straight after, post-24 h and post-48 h after a heavy bench press with a placebo vs. ibuprofen ingestion protocol. * = significant difference between these two testing times for this condition (*p* < 0.05); ^†^ = significant difference between these two testing times for both conditions (*p* < 0.05).

**Figure 2 ijerph-17-05157-f002:**
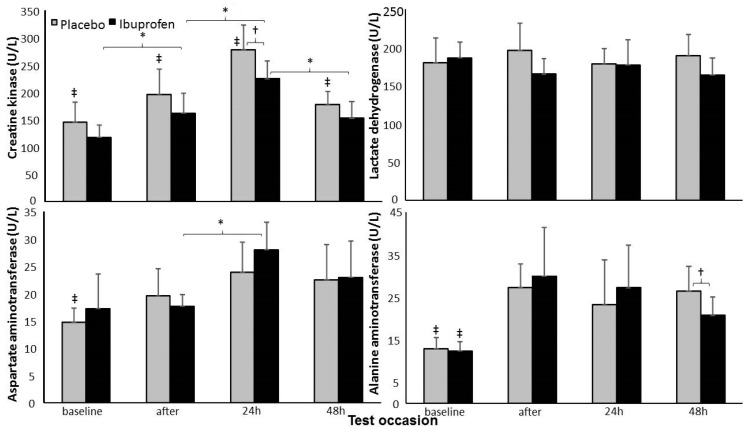
Mean (±SD) creatine kinase, lactate dehydrogenase, aspartate aminotransferase and alanine aminotransferase concentrations before, straight after, post-24 h and post-48 h after a heavy bench press with placebo vs. ibuprofen ingestion protocol. * = significant difference between these two testing times for this condition (*p* < 0.05); ^†^ = significant difference between conditions of exercise (*p* < 0.05); ^‡^ = significant difference from all other testing times for this condition (*p* < 0.05).

**Figure 3 ijerph-17-05157-f003:**
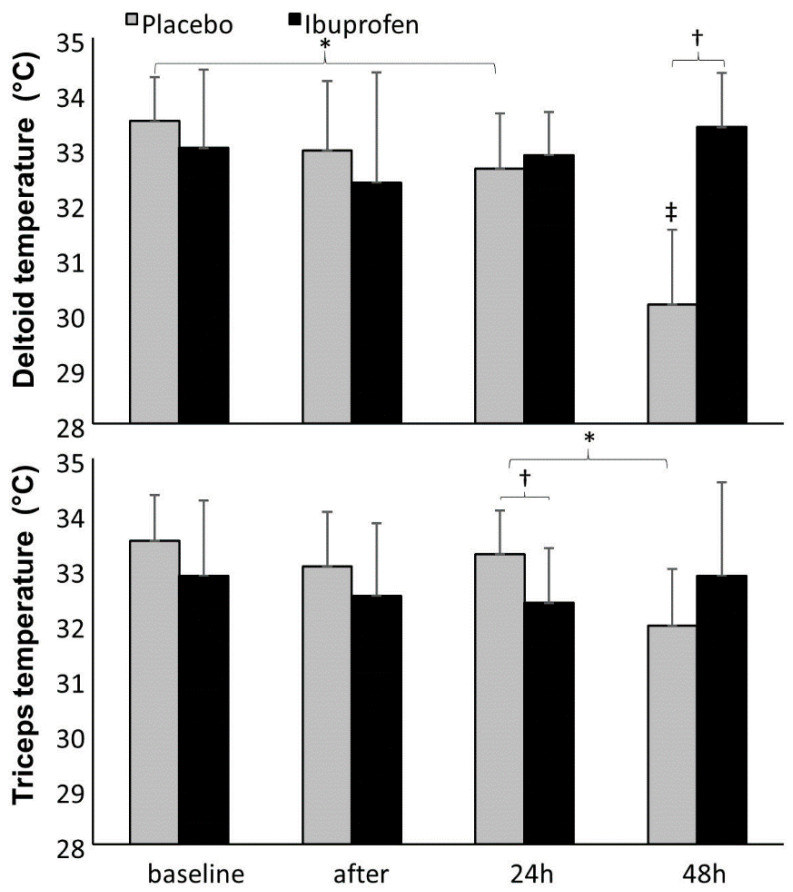
Mean (±SD) deltoid and triceps muscle temperature before, straight after, post-24 h and post-48 h after a heavy bench press with placebo vs. ibuprofen ingestion protocol. * = significant difference between these two testing times for this condition (*p* < 0.05); ^†^ = significant difference between these two conditions (*p* < 0.05); ^‡^ = significant difference with all other testing times for this condition (*p* < 0.05).

**Table 1 ijerph-17-05157-t001:** Experimental design.

Week	Day		Day	Day
Week 1	Day 11RM test		Day 21RM re-test	
Week 2(experiment)	Day 3Pre-exercisedata	Training 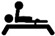	Day 3Post-exercisedata	Day 4 andFollow-up24 and 48 h
Week 3(experiment)	Day 6Pre-exercisedata	Training 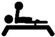	Day 7Post-exercisedata	Day 8 and 9Follow-up24 and 48 h

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
