# Peer review of "Effects of Ibuprofen Intake in Muscle Damage, Body Temperature and Muscle Power in Paralympic Powerlifting Athletes"

_ijerph, 2020, doi:10.3390/ijerph17145157_

Round 1

Reviewer 1 Report

Ibuprofen intake to reduce muscle damage, body temperature and preserve muscle power in Paralympic Powerlifting athletes

In this study Fraga et al. present results of a randomized, double-blind human trial investigating the impact of oral intake of the non-steroidal anti-inflammatory drug (NSAID) ibuprofen on muscular recovery following an acute bout of upper body resistance exercise in national elite Paralympic Powerlifting athletes. Strengths of the study include the robust trial design, the recruitment of study participants from an elite athletic population, and the use of an experimental exercise bout (bench press) with strong relevance to power lifting training and competition. The study builds on a very large existing body of literature concerning the impact of NSAID treatment on recovery of muscle force and blood markers of muscle damage over the last two decades or so, which overall unfortunately has yielded largely contradictory results. Unfortunately, I feel there are some major weaknesses in interpretations of the statistical analysis and conclusions drawn therefrom. Moreover, the writing, citations, and results presentation was at times very hard to follow. Finally, the paper entirely ignores one important aspect of the NSAID and muscle literature entirely in terms of practical implications for muscular adaptation to exercise.

Major comments:

1: One of the primary outcomes of the study was short term recovery of muscle function. For maximal isometric force there was a significant main effect of testing occasion (baseline vs. post-exercise vs. 24 h vs 48 h of recovery), no significant main effect of condition (placebo vs ibuprofen), and no significant occasion x condition interaction. There were also no apparent post-hoc differences between the two trials at any individual time-point. All of this directly conflicts with the main conclusions in the abstract that “force only decreased in the placebo condition” and the ultimate conclusion that “ibuprofen showed better results than controls in measures of muscle function”. I acknowledge that pairwise post-hoc differences over time within each trial independently appear to show a loss of force in the placebo trial and not the ibuprofen trial etc. But, really what does this mean if there was no difference between the conditions also? Many would argue that these pairwise tests split by condition should not even be performed when there is no condition x occasion interaction effect in the main two-way ANOVA and rather the post-hoc tests over time should be performed while pooling over condition (although I acknowledge it is often done in the sport science field). Overall this is unfortunately really quite weak evidence that ibuprofen is benefiting muscle force in these athletes as a primary outcome.

2: A second primary outcome of the study is serum creatine kinase (CK) concentration as a marker of muscle damage. There was a significant main effect of testing occasion and a significant main effect of condition, but no significant occasion x condition interaction effect. Most correctly interpreted what this really means that serum CK increased post-exercise similarly in both the placebo and ibuprofen conditions and that serum CK was similarly lower in the ibuprofen condition than the placebo condition irrespective of time-point. Or in other words, the placebo and ibuprofen groups were not more different post-exercise than they were pre-exercise. Since at baseline they had not yet taken the experimental treatment, the pattern for lower serum CK at baseline cannot have anything to do with ibuprofen treatment. Perhaps the post-hoc test was significant between the placebo and ibuprofen groups at 24 h post-exercise (but not at baseline)? But, even this is not clear from the presentation of the data. In any case, as point 1 above the lack of a condition x occasion interaction really suggests the ibuprofen really did not do much at all unfortunately which conflicts with the conclusions.

3: A third primary outcome claimed to be influenced by ibuprofen is muscle temperature. Unlike serum muscle force and serum CK, muscle temperature did indeed show a significant occasion x condition interaction suggesting the change over time post-exercise was indeed different between the placebo and ibuprofen trials (ibuprofen did do something here). Additionally, unlike the CK data, the muscle temperature results appear to be correctly presented. So, the biggest impact of ibuprofen in the whole study was to prevent the normal decrease in deltoid temperature post-exercise. That being said, it is unclear what the reduction in muscle temperature at 48 h post-exercise in the placebo trial indicates and thus what physiological relevance the relatively greater muscle temperature 48 h after taking ibuprofen would be. This is not discussed at all in the paper.

4: There is no mention in this paper what the long term implications of diminishing the inflammatory response to powerlifting training and/or competition in these elite Paralympic athletes may be (if ibuprofen even does this). There is a large body of literature from both human and animal studies conclusively demonstrating that NSAID treatment can interfere with muscular adaptations to mechanism loading (e.g. resistance or eccentric exercise). This does not necessarily mean that NSAID use may not be of some ergogenic benefit under certain circumstances for these athletes. But, it is not clear from the paper under what circumstances such claimed short term improvements in recovery by preserving muscle function and reducing muscle damage would be desirable in these athletes. Would the authors advise such powerlifters to routinely take ibuprofen when training? Or only in competition (perhaps to improve or maintain performance temporarily)?

5: One potential novelty of the study is their participant population which comprises national elite level Paralympic athletes and the use of an exercise stimulus presumably of strong relevancy to power lifter competitions (the bench press). However, this is never really discussed in relation to the prior literature much of which was performed in individuals unaccustomed to resistance exercise or recreational weight lifters and much of which utilized lower body exercises.

Minor comments:

Page 2, Line 50: “Have used various approaches techniques, such as”. Delete one of the extra words

Page2, Line 51: “such as cryotherapy and non-steroidal anti-inflammatory drugs (NSAIDs) (i.e. ibuprofen), both during competition and training”. Neither of the two references cited here are about cryotherapy. Please replace one of the two with a paper to support the claim of widespread use of cryotherapy

Page 2, Line 53: “Most studies have focused on recovery from aerobic activity (6)”. Perhaps this is true in say studies of hydration and carbohydrate. But, many more studies have focused on anaerobic performance (eccentric and/or resistance exercise) with regard to NSAIDs.

Page 2, Line 53: “Most studies have focused on recovery from aerobic activity (6)”. Also this reference does not really appear to be relevant to the statement about athletic recovery from aerobic activity.

Page 2, Line 54: “Peterson et al. reported that ibuprofen had no major effects on inflammatory cell concentrations, creatine kinase activity, or muscle soreness”. Peterson et al. 2003 do not report creatine kinase or muscle soreness. Please cite the relevant papers.

Page 2, Line 56: “both therapies reduced protein synthesis and prostaglandins”. Peterson et al. 2003 did not measure protein synthesis or prostaglandins at all. Please cite the relevant papers.

Page 2, Line 57: “Da silva et al. did not found any effects on fatigue, pain, lactate activity, or CL when ibuprofen was taken. Please cite the relevant papers.” This paper reports a descriptive survey of athlete use of NSAIDs and has nothing to do with the sentence above.

Page 2, Line 53-61: This is a very selective choice of three prior studies of the effect of NSAID on recovery from exercise-induced muscle damage in humans. In excess of 30 studies now exist. Obviously this is too much to cover here, but perhaps considering citing one of the many review papers on the topic?

Page2, Line 62: “Although NSAIDs are often used to alleviate exercise-induced muscle soreness”. Reference 10 found that NSAIDs had no effect on muscle pain so does not support this statement.

Page 2, Line 68: “as well as reduce de muscle damage”. Typographical error here.

Page 3, Line 99-107: How long did the training session take to complete?

Page 3, Line 108: “Participants received two capsules of IBU (400 mg)”. The exact daily dose is not clear here. Were the participants were administered two 400 mg capsules on both dosing occasions (15 min pre and 5-hour post-exercise) for a total 1600 mg daily dose. Or rather was a single 400 mg dose given at 15 min pre-exercise and 5-hour post exercise for a daily dose of 800 mg.

Page 4, Line 137: It is not clear what the EDTA plasma was used for as the primary measures presented appear to be all serum analyses?

Figure 1: Issues with the text alignment/words split over multiple lines.

Page 4: Line 147-155: “If significant effects were found a Holm-Bonerroni’s Post-Hoc was conducted”. Missing a work (test?) after Post-Hoc.

Page 4: Line 147-155: “If significant effects were found a Holm-Bonerroni’s Post-Hoc was conducted”. It is not clear from this description when various types of Post-Hoc test were used. Were pair-wise Post-Hoc tests (e.g. pre vs post-exercise for ibuprofen and placebo trials separately) when there was a trial x time interaction?

Page 4: Line 157-197: Throughout the results the word “occasion” is used for the variable of repeated tests over time within each experimental trial. I would suggest perhaps rewording this to simply “time”. This is because occasion may be confused with the two independent visits on which each of the two conditions were tested.

Page 4: Line 161:” A significant test occasion * condition interaction was found for deltoid and triceps deltoid,” Something is missing here. I assume this is referring to muscle temperature?

Page 4: Results, Line 156: “Ibuprofen digestion” used throughout. Perhaps you mean “ingestion”.

Page 5: Line 172: “The concentration (CK) levels increased significantly more after placebo digestion compared with the ibuprofen resulting in significantly higher levels after 24 h”. Increased significantly more based on what? Was the increase from baseline not significant for the ibuprofen group also? It appears it must have been significant for both groups? Also was there indeed a significant post-hoc difference between the placebo and ibuprofen trials only at 24 h (the symbols on the graph are very confusing as outlined below)? Even if there was a difference (by post-hoc test at 24 h it is at least partially due to lower levels before consuming the drug (hence no condition x occasion interaction).

Page 6: Figure 3: The symbol use is very unclear. † is stated to “denote significant difference between these two test occasions for this condition”. But, the bracket is rather within the 24 h post-exercise occasion between placebo and ibuprofen conditions. Was there indeed a post-hoc difference between the placebo and ibuprofen trials at 24 h post-exercise?

Page 6: Figure 3: Use of ‡ is also unclear for the 24 h time-point. It is meant to denote different from all other test occasions, but the bracket is between the immediate post-exercise ibuprofen and 24 h post-exercise ibuprofen occasions?

Page 6: Figure 3: Given the tendency for a possible baseline difference in CK between the two trials have the authors considered normalizing the baseline for analysis of the post-exercise effect (delta change) or analyzing the area under the curve?

Page 6: Figure 4: “† seems to be used correctly here to denote “a significant difference between these two conditions”. Why is the use and or description for this symbol different in Figure 3?

Page 6: Figure 4: “‡Use of this symbol is unclear here too however. Is it referring to the 48 h placebo condition only compared to the other gray bars? If so perhaps move it down beneath the bracket as currently it seems like it is suggesting both the gray and black 48 h post-exercise bars are different from the earlier time-points (which does not appear true for the ibuprofen group).”

Page 8: Line 243: So the authors would have hypothesized an increase in muscle temperature during recovery from exercise? How then do the authors explain that unlike in the study referenced here (27), there was no increase above baseline in muscle temperature at any time-point post-exercise in either group?

Page 8: Line 246-252: The effect of ibuprofen on deltoid temperature at 48 h post-exercise is statistically by far the most convincing finding in the entire study. However, it is not discussed at all what this might mean and of what physiological relevance it could be. Why would muscle temperature decrease during the days following exercise in the placebo condition?

Page 8: Line 254: “Little is known about the use of NSAIDs after strength training on muscle function, muscle temperature, and biomarkers of muscle damage.” This statement is simply not true in general. There are in excess of 30 studies measuring the effect of NSAIDs on muscle function and serum markers of muscle damage in human’s roughly half of which show an effect and half of which do not. I suggest the authors should focus on the novelty of their elite Paralympic Powerlifting population as I am unaware of any prior studies in this particular athletic group.

Page 7: Line 204: There is another 30 or so other papers which around half report benefits on force and half do not. Why not one of the many review papers rather than picking two of the many original papers?

Page 7: Line 222: Again many papers have measured the effect of NSAIDs on exercise induced CK, yielding essentially contradictory results. Why not cite a review? Also it is not clear why this particular athletic population would be expected to respond similarly or differently from the many studies in non-Paralympic populations

Page 7: Line 233: Not clear why inhibiting prostaglandins would impact fatigue. Prostaglandins indeed modulate pain and inflammation, but neither was measured in this study.

Page 8: Line 254: “Recovery with the use of ibuprofen in Paralympic Powerlifting athletes showed better results than controls”, This is not really supported by any of the statistics which for the most part showed no difference between the placebo and ibuprofen conditions.

Author Response

Comments and Suggestions for Authors

Ibuprofen intake to reduce muscle damage, body temperature and preserve muscle power in Paralympic Powerlifting athletes

In this study Fraga et al. present results of a randomized, double-blind human trial investigating the impact of oral intake of the non-steroidal anti-inflammatory drug (NSAID) ibuprofen on muscular recovery following an acute bout of upper body resistance exercise in national elite Paralympic Powerlifting athletes. Strengths of the study include the robust trial design, the recruitment of study participants from an elite athletic population, and the use of an experimental exercise bout (bench press) with strong relevance to power lifting training and competition. The study builds on a very large existing body of literature concerning the impact of NSAID treatment on recovery of muscle force and blood markers of muscle damage over the last two decades or so, which overall unfortunately has yielded largely contradictory results. Unfortunately, I feel there are some major weaknesses in interpretations of the statistical analysis and conclusions drawn therefrom. Moreover, the writing, citations, and results presentation was at times very hard to follow. Finally, the paper entirely ignores one important aspect of the NSAID and muscle literature entirely in terms of practical implications for muscular adaptation to exercise.

Major comments:

1: One of the primary outcomes of the study was short term recovery of muscle function. For maximal isometric force there was a significant main effect of testing occasion (baseline vs. post-exercise vs. 24 h vs 48 h of recovery), no significant main effect of condition (placebo vs ibuprofen), and no significant occasion x condition interaction. There were also no apparent post-hoc differences between the two trials at any individual time-point. All of this directly conflicts with the main conclusions in the abstract that “force only decreased in the placebo condition” and the ultimate conclusion that “ibuprofen showed better results than controls in measures of muscle function”. I acknowledge that pairwise post-hoc differences over time within each trial independently appear to show a loss of force in the placebo trial and not the ibuprofen trial etc. But, really what does this mean if there was no difference between the conditions also? Many would argue that these pairwise tests split by condition should not even be performed when there is no condition x occasion interaction effect in the main two-way ANOVA and rather the post-hoc tests over time should be performed while pooling over condition (although I acknowledge it is often done in the sport science field). Overall this is unfortunately really quite weak evidence that ibuprofen is benefiting muscle force in these athletes as a primary outcome.

Reply:We have changed the conclusion (check abstract and conclusion) to match the reviewer concern about the true lack of ibuprofen effect on muscle function. In the discussion we have also added a sentence that clearly states that there was a lack of ibuprofen effect on muscle function herein.

2: A second primary outcome of the study is serum creatine kinase (CK) concentration as a marker of muscle damage. There was a significant main effect of testing occasion and a significant main effect of condition, but no significant occasion x condition interaction effect. Most correctly interpreted what this really means that serum CK increased post-exercise similarly in both the placebo and ibuprofen conditions and that serum CK was similarly lower in the ibuprofen condition than the placebo condition irrespective of time-point. Or in other words, the placebo and ibuprofen groups were not more different post-exercise than they were pre-exercise. Since at baseline they had not yet taken the experimental treatment, the pattern for lower serum CK at baseline cannot have anything to do with ibuprofen treatment. Perhaps the post-hoc test was significant between the placebo and ibuprofen groups at 24 h post-exercise (but not at baseline)? But, even this is not clear from the presentation of the data. In any case, as point 1 above the lack of a condition x occasion interaction really suggests the ibuprofen really did not do much at all unfortunately which conflicts with the conclusions.

Reply:We have changed the text in the abstract, discussion and conclusion to address the reviewer concerns and to refrain from ill-advised conclusions.

3: A third primary outcome claimed to be influenced by ibuprofen is muscle temperature. Unlike serum muscle force and serum CK, muscle temperature did indeed show a significant occasion x condition interaction suggesting the change over time post-exercise was indeed different between the placebo and ibuprofen trials (ibuprofen did do something here). Additionally, unlike the CK data, the muscle temperature results appear to be correctly presented. So, the biggest impact of ibuprofen in the whole study was to prevent the normal decrease in deltoid temperature post-exercise. That being said, it is unclear what the reduction in muscle temperature at 48 h post-exercise in the placebo trial indicates and thus what physiological relevance the relatively greater muscle temperature 48 h after taking ibuprofen would be. This is not discussed at all in the paper.

Reply: We added in the discussion a possible explanation for these results. The ibuprofen seemed to have delayed the natural anti-inflammatory response. We also changed the abstract and the conclusions accordingly.

4: There is no mention in this paper what the long term implications of diminishing the inflammatory response to powerlifting training and/or competition in these elite Paralympic athletes may be (if ibuprofen even does this). There is a large body of literature from both human and animal studies conclusively demonstrating that NSAID treatment can interfere with muscular adaptations to mechanism loading (e.g. resistance or eccentric exercise). This does not necessarily mean that NSAID use may not be of some ergogenic benefit under certain circumstances for these athletes. But, it is not clear from the paper under what circumstances such claimed short term improvements in recovery by preserving muscle function and reducing muscle damage would be desirable in these athletes. Would the authors advise such powerlifters to routinely take ibuprofen when training? Or only in competition (perhaps to improve or maintain performance temporarily)?

Reply: Theoretically, Ibuprofen would be useful in very intense weeks of training, where the athletes perform an unusual amount of work. To use ibuprofen as means to recover form daily training does not seem clever, as NSAID prolonged use is not risk-free.

We added a sentence in the final part of the introduction to highlight this possible benefit.

5: One potential novelty of the study is their participant population which comprises national elite level Paralympic athletes and the use of an exercise stimulus presumably of strong relevancy to power lifter competitions (the bench press). However, this is never really discussed in relation to the prior literature much of which was performed in individuals unaccustomed to resistance exercise or recreational weight lifters and much of which utilized lower body exercises.

Reply: Indeed, it was not easy to confront this with previous studies in the literature due to such differences in population and methods. We understand those limitations and we added in the last part of the discussion a paragraph on this.

Minor comments:

The manuscript was changed according every of the following points raised by the reviewer. In the issues where no changes were performed or additional information was necessary we addressed below.

Page 2, Line 50: “Have used various approaches techniques, such as”. Delete one of the extra words

Reply: We agree with the reviewer. Text re-phrased.

Page2, Line 51: “such as cryotherapy and non-steroidal anti-inflammatory drugs (NSAIDs) (i.e. ibuprofen), both during competition and training”. Neither of the two references cited here are about cryotherapy. Please replace one of the two with a paper to support the claim of widespread use of cryotherapy

Reply: Reference replaced

Page 2, Line 53: “Most studies have focused on recovery from aerobic activity (6)”. Perhaps this is true in say studies of hydration and carbohydrate. But, many more studies have focused on anaerobic performance (eccentric and/or resistance exercise) with regard to NSAIDs.

Reply: We agree with the reviewer. Text re-phrased.

Page 2, Line 53: “Most studies have focused on recovery from aerobic activity (6)”. Also this reference does not really appear to be relevant to the statement about athletic recovery from aerobic activity.

Reply: Reference replaced

Page 2, Line 54: “Peterson et al. reported that ibuprofen had no major effects on inflammatory cell concentrations, creatine kinase activity, or muscle soreness”. Peterson et al. 2003 do not report creatine kinase or muscle soreness. Please cite the relevant papers.

Reply: Reference replaced

Page 2, Line 56: “both therapies reduced protein synthesis and prostaglandins”. Peterson et al. 2003 did not measure protein synthesis or prostaglandins at all. Please cite the relevant papers.

Reply: Reference replaced

Page 2, Line 57: “Da silva et al. did not found any effects on fatigue, pain, lactate activity, or CL when ibuprofen was taken. Please cite the relevant papers.” This paper reports a descriptive survey of athlete use of NSAIDs and has nothing to do with the sentence above.

Reply: Reference replaced

Page 2, Line 53-61: This is a very selective choice of three prior studies of the effect of NSAID on recovery from exercise-induced muscle damage in humans. In excess of 30 studies now exist. Obviously this is too much to cover here, but perhaps considering citing one of the many review papers on the topic?

Reply: Review paper added and paragraph rewritten

Page 2, Line 68: “as well as reduce de muscle damage”. Typographical error here.

Reply: Corrected.

Page 3, Line 99-107: How long did the training session take to complete?

Reply: 1h30min, Inserted.

Page 3, Line 108: “Participants received two capsules of IBU (400 mg)”. The exact daily dose is not clear here. Were the participants were administered two 400 mg capsules on both dosing occasions (15 min pre and 5-hour post-exercise) for a total 1600 mg daily dose. Or rather was a single 400 mg dose given at 15 min pre-exercise and 5-hour post exercise for a daily dose of 800 mg.

Reply: 2x 400mg. Text re-phrased.

Page 4, Line 137: It is not clear what the EDTA plasma was used for as the primary measures presented appear to be all serum analyses?

Reply: Corrected. All were serum analyses.

Figure 1: Issues with the text alignment/words split over multiple lines.

Reply: Corrected

Page 4: Line 147-155: “If significant effects were found a Holm-Bonerroni’s Post-Hoc was conducted”. Missing a work (test?) after Post-Hoc.

Reply: Corrected

Page 4: Line 147-155: “If significant effects were found a Holm-Bonerroni’s Post-Hoc was conducted”. It is not clear from this description when various types of Post-Hoc test were used. Were pair-wise Post-Hoc tests (e.g. pre vs post-exercise for ibuprofen and placebo trials separately) when there was a trial x time interaction?

Reply: We agree with the reviewer. Text re-phrased.

Page 4: Line 157-197: Throughout the results the word “occasion” is used for the variable of repeated tests over time within each experimental trial. I would suggest perhaps rewording this to simply “time”. This is because occasion may be confused with the two independent visits on which each of the two conditions were tested.

Reply: Corrected

Page 4: Line 161:” A significant test occasion * condition interaction was found for deltoid and triceps deltoid,” Something is missing here. I assume this is referring to muscle temperature?

Reply: Corrected

Page 4: Results, Line 156: “Ibuprofen digestion” used throughout. Perhaps you mean “ingestion”.

Reply: Corrected

Page 5: Line 172: “The concentration (CK) levels increased significantly more after placebo digestion compared with the ibuprofen resulting in significantly higher levels after 24 h”. Increased significantly more based on what? Was the increase from baseline not significant for the ibuprofen group also? It appears it must have been significant for both groups? Also was there indeed a significant post-hoc difference between the placebo and ibuprofen trials only at 24 h (the symbols on the graph are very confusing as outlined below)? Even if there was a difference (by post-hoc test at 24 h it is at least partially due to lower levels before consuming the drug (hence no condition x occasion interaction).

Reply: We agree with the reviewer. Text re-phrased.

Page 6: Figure 3: The symbol use is very unclear. † is stated to “denote significant difference between these two test occasions for this condition”. But, the bracket is rather within the 24 h post-exercise occasion between placebo and ibuprofen conditions. Was there indeed a post-hoc difference between the placebo and ibuprofen trials at 24 h post-exercise?

Reply: Corrected

Page 6: Figure 3: Use of ‡ is also unclear for the 24 h time-point. It is meant to denote different from all other test occasions, but the bracket is between the immediate post-exercise ibuprofen and 24 h post-exercise ibuprofen occasions?

Reply: Corrected. Symbol was moved into the right place.

Page 6: Figure 3: Given the tendency for a possible baseline difference in CK between the two trials have the authors considered normalizing the baseline for analysis of the post-exercise effect (delta change) or analyzing the area under the curve?

Reply: Indeed, we have not.

Page 6: Figure 4: “† seems to be used correctly here to denote “a significant difference between these two conditions”. Why is the use and or description for this symbol different in Figure 3?

Reply: Corrected

Page 6: Figure 4: “‡Use of this symbol is unclear here too however. Is it referring to the 48 h placebo condition only compared to the other gray bars? If so perhaps move it down beneath the bracket as currently it seems like it is suggesting both the gray and black 48 h post-exercise bars are different from the earlier time-points (which does not appear true for the ibuprofen group).”

Reply: Corrected. Symbol was moved into the right place.

Page 8: Line 243: So the authors would have hypothesized an increase in muscle temperature during recovery from exercise? How then do the authors explain that unlike in the study referenced here (27), there was no increase above baseline in muscle temperature at any time-point post-exercise in either group?

Reply: Baseline measurement of temperature may have influenced these results. We added more comprehensive explanation on this in the discussion section,

Page 8: Line 246-252: The effect of ibuprofen on deltoid temperature at 48 h post-exercise is statistically by far the most convincing finding in the entire study. However, it is not discussed at all what this might mean and of what physiological relevance it could be. Why would muscle temperature decrease during the days following exercise in the placebo condition?

Reply: A possible explanation was included in the discussion.

Page 8: Line 254: “Little is known about the use of NSAIDs after strength training on muscle function, muscle temperature, and biomarkers of muscle damage.” This statement is simply not true in general. There are in excess of 30 studies measuring the effect of NSAIDs on muscle function and serum markers of muscle damage in human’s roughly half of which show an effect and half of which do not. I suggest the authors should focus on the novelty of their elite Paralympic Powerlifting population as I am unaware of any prior studies in this particular athletic group.

Reply: We agree with the reviewer. Text re-phrased.

Page 7: Line 204: There is another 30 or so other papers which around half report benefits on force and half do not. Why not one of the many review papers rather than picking two of the many original papers?

Reply: One of the studies compared herein was chosen because of the same study design and ibuprofen administration (though in running exercise). The other study was chosen because it reports the same movement as that in the current study, bench press.

Page 7: Line 222: Again many papers have measured the effect of NSAIDs on exercise induced CK, yielding essentially contradictory results. Why not cite a review? Also it is not clear why this particular athletic population would be expected to respond similarly or differently from the many studies in non-Paralympic populations

Reply: Paragraph was rewritten and a review study added.

Page 7: Line 233: Not clear why inhibiting prostaglandins would impact fatigue. Prostaglandins indeed modulate pain and inflammation, but neither was measured in this study.

Reply: Fatigue was removed from the paragraph.

Page 8: Line 254: “Recovery with the use of ibuprofen in Paralympic Powerlifting athletes showed better results than controls”, This is not really supported by any of the statistics which for the most part showed no difference between the placebo and ibuprofen conditions.

Reply: We agree with the reviewer. Text re-phrased.

Reviewer 2 Report

The aim of this study was to evaluate the effect of ingesting Ibuprofen on the post-workout recovery of muscle damage, body temperature, and muscle power indicators in Paralympic Powerlifting athletes. I think that paper is very important, it Conclusion to The intake of non-steroidal anti-inflammatory drugs (NSAIDs), such as ibuprofen, has been used by athletes to relieve the symptoms of muscle injury and pain. However, little is known about the use of NSAIDs after strength training on muscle function, muscle temperature, and biomarkers of muscle injury. In the present study, it was found that recovery with the use of ibuprofen in Paralympic Powerlifting athletes showed better results than controls, in relation to measures of muscle function and blood markers of muscle damage.

Author Response

The aim of this study was to evaluate the effect of ingesting Ibuprofen on the post-workout recovery of muscle damage, body temperature, and muscle power indicators in Paralympic Powerlifting athletes. I think that paper is very important, it Conclusion to The intake of non-steroidal anti-inflammatory drugs (NSAIDs), such as ibuprofen, has been used by athletes to relieve the symptoms of muscle injury and pain. However, little is known about the use of NSAIDs after strength training on muscle function, muscle temperature, and biomarkers of muscle injury. In the present study, it was found that recovery with the use of ibuprofen in Paralympic Powerlifting athletes showed better results than controls, in relation to measures of muscle function and blood markers of muscle damage.

Reply: We have carefully checked the comments inserted by the reviewer in the document that was attached and we have performed the necessary changes in the manuscript.

Reviewer 3 Report

In this manuscript, Fraga and colleagues report on the acute effects of 400mg ibuprofen intake on isometric force development, serum CK levels and muscle temperature in Paralympic athletes after 1RM test intervention. I think the paper is straightforward and reports interesting trends of ibuprofen-associated effects. I would kindly ask the Authors to address the following points:

-  the title should be changed to something like “effects of ibuprofen on muscle damage, …” to maintain a more impartial tone;

- were there significant differences in 1RM weight, body mass, training length in placebo vs ibuprofen groups at baseline? Also, I assume all participants were male, is that correct? if so, this should be clearly stated. If not, analyses for sex-specific differences for all reported results (force, CK, temperature) should be performed;

- single data points should be shown on top of all histogram bars;

- authors should emphasize the relevance of the muscle temperature findings and - specifically - of the significant recovery/increase in muscle temperature with ibuprofen.

Author Response

We appreciate the reviewer’s comments and we have attempted to address them, as follows:

In this manuscript, Fraga and colleagues report on the acute effects of 400mg ibuprofen intake on isometric force development, serum CK levels and muscle temperature in Paralympic athletes after 1RM test intervention. I think the paper is straightforward and reports interesting trends of ibuprofen-associated effects. I would kindly ask the Authors to address the following points:

-  the title should be changed to something like “effects of ibuprofen on muscle damage, …” to maintain a more impartial tone;

Reply: Title was changed

- were there significant differences in 1RM weight, body mass, training length in placebo vs ibuprofen groups at baseline? Also, I assume all participants were male, is that correct? if so, this should be clearly stated. If not, analyses for sex-specific differences for all reported results (force, CK, temperature) should be performed;

Reply:All participants were males and this was a cross-over design. Hence, all the participants performed the experiment under the two exercise conditions.

- single data points should be shown on top of all histogram bars;

Reply:We are sorry but we may not have fully understand the reviewer request.

Does your suggestion refer to the mean and (or) to the SD values? Is the inclusion of a gridline helpful in solving this?

- authors should emphasize the relevance of the muscle temperature findings and - specifically - of the significant recovery/increase in muscle temperature with ibuprofen.

Reply: In the abstract, the results and in the discussion section, text was changed to highlight the increased temperature with Ibuprofen.

Round 2

Reviewer 1 Report

Ibuprofen intake to reduce muscle damage, body temperature and preserve muscle power in Paralympic Powerlifting athletes

I thank the authors for considering my suggestions to further improve their manuscript.

The authors have addressed my major comments satisfactorily in my opinion and I believe the manuscript has overall improved greatly since the initial submission.

In my opinion this study is a worthwhile addition to the literature concerning the effects of NSAIDs on exercise recovery due to the strength of the study design and the novelty of the elite athletic population in question.

Minor comments:

Page 1, Line 32: “Temperature rose” change to “Muscle temperature rose” or specify which muscle (if not both).

Page 1, Line 32: “Temperature rose significant”. Please change to “significantly”.

Page 2, Line 50: “To speed recovery” perhaps “To speed up recovery”?

Page 3, Line 54: The focus of the review article cited here (reference 5) is indeed aerobic activity. But, subsequent references 6 and 7 tested resistance exercise and maximal eccentric exercise respectively. Perhaps this could be made a bit clearer so the reader can appreciate the opposing findings to Tokmakisis et al. even with similar exercise models.

Page 5, Line 171: “Creatine kinase concentration levels increased significantly to 24h after the exercise”. Please change to “Creatine kinase concentration levels increased significantly at 24h after the exercise.”

Page 7, Line 208: “Contrarily, Correa et al. (19) did not found significant”. Please change to “Contrarily, Correa et al. (19) did not find significant”.

Page 8, Line 252: “These temperature increases at 24-h and 48 h post-exercise may reflect, at least in the ibuprofen condition, a delay in the anti-inflammatory”. Something is missing here at the end of the sentence. Perhaps “A delay in the anti-inflammatory response to resistance exercise?”

Page 8, Line 269: “Ibuprofen use seems to have cause a delay in the anti-inflammatory response following exercise”. I think this last sentence could be a bit clearer on what basis the authors are concluding this. Perhaps something like “Furthermore, the effect of ibuprofen to prevent the normal decrease in muscle temperature during post-exercise recovery may potentially be indicative of a delay in the anti-inflammatory response”?

Author Response

I thank the authors for considering my suggestions to further improve their manuscript.

The authors have addressed my major comments satisfactorily in my opinion and I believe the manuscript has overall improved greatly since the initial submission.

In my opinion this study is a worthwhile addition to the literature concerning the effects of NSAIDs on exercise recovery due to the strength of the study design and the novelty of the elite athletic population in question.

Minor comments:

Page 1, Line 32: “Temperature rose” change to “Muscle temperature rose” or specify which muscle (if not both).

 Reply: Re-phrased. There was a mistake in the significant results highlighted here.

Page 1, Line 32: “Temperature rose significant”. Please change to “significantly”.

Reply: Corrected.

Page 2, Line 50: “To speed recovery” perhaps “To speed up recovery”?

Reply: Corrected.

Page 3, Line 54: The focus of the review article cited here (reference 5) is indeed aerobic activity. But, subsequent references 6 and 7 tested resistance exercise and maximal eccentric exercise respectively. Perhaps this could be made a bit clearer so the reader can appreciate the opposing findings to Tokmakisis et al. even with similar exercise models.

Reply: Re-phrased.

Page 5, Line 171: “Creatine kinase concentration levels increased significantly to 24h after the exercise”. Please change to “Creatine kinase concentration levels increased significantly at 24h after the exercise.”

Reply: Corrected.

Page 7, Line 208: “Contrarily, Correa et al. (19) did not found significant”. Please change to “Contrarily, Correa et al. (19) did not find significant”.

Reply: Corrected.

Page 8, Line 252: “These temperature increases at 24-h and 48 h post-exercise may reflect, at least in the ibuprofen condition, a delay in the anti-inflammatory”. Something is missing here at the end of the sentence. Perhaps “A delay in the anti-inflammatory response to resistance exercise?”

Reply: Corrected.

Page 8, Line 269: “Ibuprofen use seems to have cause a delay in the anti-inflammatory response following exercise”. I think this last sentence could be a bit clearer on what basis the authors are concluding this. Perhaps something like “Furthermore, the effect of ibuprofen to prevent the normal decrease in muscle temperature during post-exercise recovery may potentially be indicative of a delay in the anti-inflammatory response”?

Reply: Corrected.